# The Impact of Coffee and Its Selected Bioactive Compounds on the Development and Progression of Colorectal Cancer In Vivo and In Vitro

**DOI:** 10.3390/molecules23123309

**Published:** 2018-12-13

**Authors:** Rafał J. Bułdak, Tomasz Hejmo, Marcin Osowski, Łukasz Bułdak, Michał Kukla, Renata Polaniak, Ewa Birkner

**Affiliations:** 1Department of Biochemistry, School of Medicine with the Division of Dentistry in Zabrze, Medical University of Silesia, 41-808 Zabrze, Poland; tomekhejmo@gmail.com (T.H.); ebirkner@sum.edu.pl (E.B.); 2Department of Human Nutrition, School of Public Health in Bytom, Medical University of Silesia, 41-902 Bytom, Poland; marcin_osowski@poczta.fm (M.O.); rpolaniak@sum.edu.pl (R.P.); 3Department of Internal Medicine and Clinical Pharmacology, School of Medicine in Katowice, Medical University of Silesia, 40-752 Katowice, Poland; lbuldak@gmail.com; 4Department of Gastroenterology and Hepatology, School of Medicine in Katowice, Medical University of Silesia, 40-752 Katowice, Poland; kuklamich@poczta.onet.pl

**Keywords:** colorectal cancer, coffee, caffeine, caffeic acid, chlorogenic acid, kahweol, cancer progression

## Abstract

Coffee is one of the most popular beverages worldwide. Coffee contains bioactive compounds that affect the human body such as caffeine, caffeic acid, chlorogenic acids, trigonelline, diterpenes, and melanoidins. Some of them have demonstrated potential anticarcinogenic effects in animal models and in human cell cultures, and may play a protective role against colorectal cancer. Colorectal cancer (CRC) is the third leading cause of cancer-related mortality in the USA and other countries. Dietary patterns, as well as the consumption of beverages, may reduce the risk of CRC incidence. In this review, we focus on published epidemiological studies concerning the association of coffee consumption and the risk of development of colorectal cancer, and provide a description of selected biologically active compounds in coffee that have been investigated as potential cancer-combating compounds: Caffeine, caffeic acid (CA), chlorogenic acids (CGAs), and kahweol in relation to colorectal cancer progression in in vitro settings. We review the impact of these substances on proliferation, viability, invasiveness, and metastasis, as well as on susceptibility to chemo- and radiotherapy of colorectal cancer cell lines cultured in vitro.

## 1. Introduction

Coffee is one of the most popular beverages in the world. Its consumption in Europe has been increasing, and currently it averages around 4.1 kg/person/year. The highest intake of coffee per capita has been observed in Finland (12 kg/year) [1]. Coffee contains many substances that affect the human body, which include polyphenols, among them caffeine, caffeic acid, trigonelline, chlorogenic acid, and diterpenes such as cafestol and kahweol [1]. Several research teams have demonstrated the anti-inflammatory and anticarcinogenic potential of these components [1]. Additionally, in many observational studies, coffee consumption has been associated with a reduced risk of several types of cancer [2], diabetes mellitus type 2, ischemic stroke, Alzheimer’s disease, and Parkinson’s disease [3]. Many research reports have linked the reduced risk of occurrence of the aforementioned diseases with polyphenols in the diet [4,5,6,7,8]. High coffee consumption is positively correlated with smoking, the number of cigarettes smoked per day, as well as with meat consumption [9].

Coffee has a great impact on the human body as a whole. Coffee-derived bioactive compounds, especially caffeine, chlorogenic acid, and caffeic acid, are present in serum and urine. Meta-analysis and cohort studies have revealed beneficial, rather than harmful, effects of coffee consumption on people’s health [10,11]. People consuming 3–4 cups per day demonstrated a reduced risk of all causes of cardiovascular mortality, as well as of the incidence of cardiovascular diseases, while high consumption of coffee was related to a lower risk of incidence of specific cancers and metabolic, digestive, liver, or neurological conditions. Some studies have reported that habitual consumption of coffee reduces the concentration of inflammation markers, such as interleukin-18, E-selectin, C-reactive protein, adipokines, or IFN-γ, and increases the concentration of adiponectin in serum, which generates anti-inflammatory effects and increases sensitivity to insulin [12,13,14,15], while others have revealed increased levels of C-reactive protein, interleukin-6, interleukin-10, TNF-α, E-selectin, and plasma fibrinogen [16,17]. On the other hand, systematic review by Paiva et al. showed the absence of changes in C-reactive protein concentration after coffee consumption and decreased concentration after caffeine intake [17]. Changes in the expression of genes, especially those involved in inflammation in human, animal, and in vitro models [18,19,20], were reported. According to one of the animal studies, bean roasting affected the anti-inflammatory and antioxidant properties of coffee [21]. Coffee has a variable impact on serum cholesterol level, as increased or unaffected concentrations have been reported [12,16,22,23]. A study by Andersen et al. showed that consumption of coffee was associated with reduced risk of death related to inflammatory and cardiovascular diseases in postmenopausal women [24]. According to recent meta-analysis, the consumption of caffeinated and decaffeinated coffee may significantly reduce the incidence rate of type 2 diabetes mellitus [25,26]: However, high consumption of coffee by patients with type 1 diabetes was related to the metabolic syndrome [27].

Chuang et al. identified DNA methylation (DNAm) sites in human genomes that relate to habitual coffee consumption [28]. DNAm has been proposed as a potential epigenetic mediator for caffeine’s influence on health. The authors found CpG sites located near 11 genes associated with habitual coffee consumption in peripheral blood mononuclear cells. Additionally, many differentially methylated CpGs were located in or near genes known to be associated with coffee-related diseases. Authors have hypothesized that this may be the possible mechanism of the beneficial effect of coffee.

Animal studies conducted by Nakayama and Oishi showed decreased counts of *Escherichia coli* in the distal small intestine and proximal colon in mice consuming coffee [29]. The authors also reported decreased counts of *Enterococcus* spp., *Bacteroides* spp., and *Clostridium* spp., but an increased count of *Lactobacillus* spp. in the proximal colon, when compared to mice drinking sterile water. The addition of coffee to agar inhibited the growth of *E. coli* and *Enterococcus faecalis*. In a study performed by Jaquet et el., it was shown that coffee increased the count of *Bifidobacterium* spp., which is known to be a beneficial health factor and is reknowned for its metabolic activity [30].

Coffee consumption has an impact on gut microbiota [31]. A study by Cuervo et al. showed that *Clostridium*, *Lactococcus*, and *Lactobacillus* spp. were more abundant in subjects diagnosed with rhinitis and allergic asthma, which resulted from phenolic acids from coffee.

The content of bioactive compounds in coffee depends on many conditions [32,33]. The concentration of caffeine in coffee varies, depending on the method of preparation and coffee type (72–130 mg in ~240 mL of brewed coffee and 58–76 mg in a single shot of espresso), bean roasting, and procedures applied by baristas (152–232 mg/serving) [34,35]. Additionally, the boiling water pouring number and coffee ground size influence the concentration of chlorogenic acid content (75–306 mg/serving) [33,35], while the concentration of kahweol depends on the potential usage of paper filter and the characteristics of such [14].

The above-mentioned data inspired the authors to investigate more thoroughly the specific compounds contained in coffee and to study their influence on colorectal cancer progression. This review paper focuses on the current state of knowledge about the impact of coffee consumption on colorectal cancer (CRC) development in epidemiological studies and on major bioactive compounds of coffee that have been investigated in in vitro studies concerning colorectal cancer: Caffeine, caffeic acid (CA), chlorogenic acids (CGAs), and kahweol (Figure 1), and their influence on CRC cell progression. We review the impact of these substances on proliferation, viability, invasiveness, and metastatic potential and properties, as well as the sensitivity of CRC cell lines to chemo- or radiotherapy.

Colorectal cancer is the third leading cause of cancer-related mortality in the United States (US). The estimated five-year survival rates range from approximately 90% for patients with stage I disease to less than 5% for patients with stage IV disease. The causes of CRC are complex and involve the interaction between genetic and environmental factors. The vast majority of CRC cases can be attributed to environmental causes, as they account for more than 70% of all incidences [36]. Some aspects of western dietary patterns may also influence the risk of CRC. Adiposity and higher levels of insulin in obese subjects are connected with an increased risk of CRC [37]. Diets rich in animal products and processed grains have been associated with higher C-peptide concentrations and elevated insulin levels in the serum of patients [38,39,40], whereas diets rich in whole grains, fiber, fruits, vegetables, as well as the consumption of coffee have typically been connected with lower concentrations of insulin or C-peptide [41,42,43,44]. Fung et al. demonstrated that specific dietary patterns were correlated with elevated C peptide concentrations and an increased incidence of colon cancer, especially in women who were overweight or whose lifestyle was sedentary [37].

Caffeine, caffeic acid, chlorogenic acids, and kahweol have shown potential anticarcinogenic effects in animal models and human cell cultures and may play a protective role against CRC [9]. Results of epidemiological studies have provided different results. Most of them have revealed no correlation between coffee consumption and CRC risk, some have shown decreased risk, and only a few have shown increased risk. A recent meta-analysis reported a reduced incidence of colon cancer in high versus no- or low-coffee-consuming subjects in case–control studies, but not in cohort studies for CRC [2]. The epidemiological assessment of the influence of coffee on colorectal cancer risk and progression has been limited to general coffee consumption and caffeine intake, while in vitro studies are more precise, focusing on single bioactive compounds present in coffee. The results of in vitro study data have suggested that all investigated bioactive compounds may prevent or reduce the growth of colorectal cancer cells. However, it is too early to recommend them for therapeutic application for the prevention of cancer, on the basis of the available evidence.

## 2. Coffee and Colorectal Cancer In Vivo

Chemoprotection against oxidative stress and DNA damage by means of coffee has been reported in in vitro research and in studies on rats and mice [45,46,47,48,49]. Soares et al. treated rats with pure caffeine, and both caffeinated and decaffeinated coffee, after exposure to carcinogens [49]. They found a decreased development of pre-neoplastic lesions in the colon only in rats treated with caffeinated coffee. Anti-metallothionein antibody staining revealed that decaffeinated coffee lacked a chemoprotective effect, which was induced by caffeinated coffee, and an analysis of COX-2 expression revealed that only caffeine and caffeinated coffee diminished carcinogen-related inflammatory reaction. On the other hand, caffeine and caffeinated and decaffeinated coffee reduced DNA damage in the colon, which was generated by carcinogens. According to those results, the authors claimed that coffee as a beverage had higher chemoprotective potential than caffeine alone. Kang et al. assessed the impact of caffeic acid, chlorogenic acid, and decaffeinated coffee on metastatic potential in tumor-bearing mice [50]. Mice were grafted with CT-26 colon cancer cells and then treated with different concentrations of compounds. After two weeks, the volume of lung tumors was measured. Decaffeinated coffee and CGA suppressed lung metastases in a dose-dependent manner, resulting in a decreased number and volume of tumor nodules. Additionally, CA decreased tumor nodule numbers and their weight. Both decaffeinated coffee and CGA inhibited COX-2, MMP-2, and MMP-9 expression, as well as ERK phosphorylation. Decaffeinated coffee and CA decreased MEK1 activity, while TOPK activity was inhibited by coffee, CA, and CGA, to a greater extent by CA than CGA. This resulted in the inhibition of metastases formation and enlargement.

Many epidemiological studies exploring the association between coffee consumption and colorectal cancer risk have been published. Their authors have provided much additional information about patients, such as information concerning sex, age, physical activity, dietary habits, smoking, alcohol consumption, education level, and others. On the other hand, coffee consumption is most commonly measured by the number of cups consumed per day. Some authors have distinguished caffeinated and decaffeinated coffee, but have provided no information about bean roasting level, brewing method, the addition of milk, sugar, or alcohol, or the amount of coffee used to brew one cup, nor have they distinguished coffee brewed from *Coffea arabica* and *Coffea canephora* (robusta) beans, which are very different in terms of the concentration of bioactive compounds, especially caffeine and kahweol [51]. Providing such information is very difficult, but without it the results of epidemiological studies cannot be compared and do not provide clear answers. This may serve as an explanation for the inconsistency between the observed inhibitory effect of bioactive compounds from coffee on colorectal cancer cells in the in vitro studies and inconclusive associations between coffee consumption and colorectal cancer risk in vivo.

Serum biomarkers related to habitual coffee consumption may help in understanding the association between coffee intake and colorectal cancer risk. A study by Guertin et al. showed that certain serum metabolites allow for distinguishing coffee drinkers from abstainers [52]. Additionally, higher concentrations of theophylline, caffeine, and paraxanthine were significantly inversely associated with colorectal cancer.

No significant association between total (not distinguishing between caffeinated and decaffeinated coffee), caffeinated, and decaffeinated coffee consumption and total colorectal cancer incidence risk was found in the prospective cohort study by Dik et al. [9]. Similar results were observed for caffeinated coffee (hazard ratio (HR) 1.18, *p*-trend 0.15). A prospective cohort study by Dominianni et al. also found no significant association between coffee consumption and the risk of colorectal cancer, also concerning the subsite distribution of cancers: Proximal, distal, or rectal cancer [53]. Bidel et al. investigated the potential association between coffee consumption and CRC incidence risk in a prospective cohort study conducted in Finland, the country with the highest consumption of coffee in the world [54]. The authors analyzed a wide range of consumption patterns, from 1 to 10 cups per day. No relationship between ingestion of coffee and incidence of colorectal, colon, and rectal cancer was found. Likewise, no association was found after adjusting patients for sex. According to the cohort study by Lukic et al., the association between coffee consumption and CRC risk was not significant (*p*-trend 0.10) [55]. The volume of one cup was estimated as 2.1 dL. Other population-based prospective cohort studies of Swedish patients have been performed by Terry et al., Mucci et al., Larsson et al., and Nilsson et al. [56,57,58,59]. No correlations between coffee consumption and colorectal, colon, proximal colon, distal colon, or rectal cancer were found. No associations of caffeinated coffee consumption and the incidence rate of colorectal, colon, and rectal cancer were found in a study by Michels et al. for groups pooled and adjusted for sex [60]. Nevertheless, the consumption of decaffeinated coffee was related to a marginally lower risk of colorectal and rectal cancer in pooled groups (*p*-trend 0.08, *p*-trend 0.06, respectively). No association was found also in Japanese patients with regard to colorectal, proximal, and distal colon cancer [61]. No association was observed by Peterson et al. in a prospective study for colon and rectal cancers, either [62].

A population-based case–control study by Schmit et al. found a significant association between coffee consumption and decreased risk of colorectal cancer [63]. Coffee drinkers turned out to demonstrate a lower probability of developing colorectal, colon, and rectal cancer than nondrinkers (odds ratio (OR) 0.74, *p* < 0.001; OR 0.74, *p* < 0.001; OR 78, *p* < 0.04, respectively). Lower risk of CRC was significant also for decaffeinated coffee (OR 0.82, *p* < 0.001). Stratifying the group according to daily consumption of coffee and subsite-specific cancers allowed the authors to observe decreasing chances of developing colorectal, colon, and rectal cancer in a dose-dependent manner. The authors also distinguished different types of brewing procedures. Only boiled coffee was significantly associated with a lower risk of CRC (OR 0.82, *p* < 0.004), while the associations of espresso, instant, and filtered coffee were not significant. The authors explained the anticancer potential of unfiltered, boiled coffee with a high content of cafestol and kahweol, which may exert anticancerogenic activity and are removed from coffee during filtering. Similar associations between coffee consumption and a reduced risk of CRC was also found by Sinha et al. [64]. The risk of colorectal, colon, and proximal colon cancers significantly decreased in a dose-dependent manner for caffeinated and decaffeinated coffee alike. The highest decrease of CRC and colon cancer risk was observed in patients who drank more than 6 cups of decaffeinated coffee per day (HR 0.74, *p*-trend < 0.001; HR 0.73, *p*-trend 0.005, respectively), while the lowest risk of proximal colon was observed in patients who drank more than 6 cups of caffeine-containing coffee per day (HR 0.59, *p* < 0.001). The consumption of decaffeinated coffee was significantly associated with a reduced risk of rectal cancer (HR 0.71 for 4–5 cups/day, *p*-trend 0.003). There were no significant associations between consumption of all types of coffee and distal colon cancer as well as caffeinated coffee consumption and rectal cancer.

Coffee consumption was found to be significantly associated with a reduced risk of CRC in Japanese people [65]. The risk of colorectal, colon, and distal colon cancer decreased with more frequent consumption of coffee, up to 1–3 cups per day, but increased in people who drank more than 4 cups per day. There was no association noted for proximal colon cancer or rectal cancer. According to the results of another population-based prospective cohort study of Japanese men and women, the consumption of coffee was not associated with CRC risk for both men and women, but reduced the risk of invasive colon cancer in women ingesting less than 1, 1–2, and more than 3 cups of coffee per day (relative risk (RR) 0.79 vs. 0.74 vs. 0.44, respectively, *p*-trend 0.04) [66].

Coffee consumption was associated with decreased colorectal cancer mortality in women in a study conducted by Sugiyama et al. [67]. The ingestion of 1 cup per day reduced the risk of CRC more than merely occasional consumption (HR 0.26 vs. 0.74, *p*-trend 0.02). That relationship was not observed in men. In opposition to results obtained by Naganuma et al., a study by Nakagawa et al. found a significant inverse association between coffee consumption and the risk of CRC, with the highest reduction for the consumption of ≥3 cups per day (OR 0.78, *p*-trend 0.009) [68]. The authors assessed two case–control groups, and the results were calculated for both together. The authors also found a significant dose-dependent inverse risk of developing distal colon cancer (OR 0.80 for <1 cup/day, OR 0.78 for 1–2 cups/day, OR 0.77 for ≥3 cups per day, *p*-trend 0.048), while there was no association for proximal colon or rectal cancers. The odds ratio for CRC remained significant only for men after adjusting to sex (*p*-trend 0.033).

The most interesting results were obtained in a case–control study by Azzeh et al. [69]. The study, conducted on 137 patients diagnosed with colorectal cancer and 164 healthy controls, revealed that the consumption of coffee significantly reduced the risk of CRC, even by 92%. The risk reduction was not dose-dependent, and the lowest risk was observed for the consumption of 1–2 and more than 5 cups per day (OR 0.08; OR 0.11, *p*-trend 0.002). The patients from the control group consumed more coffee than patients with CRC (*p* < 0.001). There was no significant difference in the consumption of red meat and processed meat between the groups. Red and processed meat consumption was reported to increase the risk of colorectal cancer [70,71,72,73]. Also, the incidence rate of CRC in the Saudi population was high [74].

On the other hand, the study by Groessl et al. revealed the dose-dependent increased risk of colorectal cancer incidence in women [75]. When compared to the non-coffee-drinking control, the colorectal cancer hazard ratio was 1.14 for women drinking up to 4 cups of caffeinated coffee per day and 1.15 for those drinking more than 4 cups per day (*p*-trend 0.04): However, after multivariate adjustment, the association remained significant, but the confidence interval for patients drinking more than 4 cups per day was of borderline insignificance. Additionally, the authors distinguished between colon, rectum, and rectosigmoid cancers, and calculated HR for women drinking up to 4 and more than 4 cups of caffeinated coffee per day (HR 1.11, HR 1.17, *p*-trend 0.07; HR 1.30, HR 0.95, *p*-trend 0.53; HR 1.15, HR 1.39, *p*-trend 0.40, respectively). The authors constructed survival rate (Kaplan-Meier) curves, which showed lower survival free of colorectal cancer for patients with higher coffee intake.

A case-control study by Green et al. revealed that consumption of <1 cup/week and ≥1 cup/week of iced coffee significantly increased the risk of colorectal cancer, while consumption of <1 cup/month decreased it (adjusted odds ratio (AOR) 1.64, AOR 1.19, AOR 0.89, respectively, *p*-trend 0.035) [76]. The consumption of hot and decaffeinated coffee was not associated with CRC. As the authors admitted, the results of their study were inconsistent with other reports, which may have been caused by different methods of preparing coffee in Australia and in Europe, where most case–control studies have been performed. Green et al. observed that consumption of hot coffee was associated with elevated risk of distal colon cancer for the whole range of consumption patterns (AOR 1.55 for <1 cup/day, AOR 1.91 for 1 cup/day, AOR 1.78 for ≥2 cups/day, *p*-trend 0.085). Additionally, the consumption of <1 cup/week and ≥1 cup/week of iced coffee was associated with significantly greater risk of rectal cancer (AOR 2.06, AOR 1.52, respectively, *p*-trend 0.004). The authors pointed out that iced coffee in Australia is pre-made with milk and sugar and presumed that additional sugar increases the risk of rectal cancer.

Yamada et al. found a statistically significant association between the consumption of coffee and the incidence of colorectal cancer in Japanese male patients in a prospective cohort study [77]. Men who consumed 2–3 cups or more than 4 cups of coffee per day faced a significantly higher risk of colorectal cancer (HR 1.21, HR 1.57, respectively, *p*-trend 0.03). Interestingly, this association was not significant in women (*p*-trend 0.61). The authors additionally distinguished subsite-related cancers. The association remained significant in both groups of men for colon cancer but not for rectal cancer (HR 1.26, *p*-trend 0.03; HR 1.79, *p*-trend 0.53, respectively).

Hu et al. and Guercio et al. conducted a prospective study on patients diagnosed with CRC and colon cancer. Both studies revealed an inverse association between post-diagnostic coffee consumption and CRC-related as well as overall mortality. Guercio et al. analyzed patients with stage III colon cancer during chemotherapy [78]. Patients consuming 4 or more cups of total and caffeinated coffee per day had significantly lower hazard ratios for colon cancer recurrence or mortality risk (HR 0.58, *p*-trend 0.002; HR 0.48, *p*-trend 0.002, respectively) when compared to non-coffee-drinkers. Additionally, increasing caffeine intake was related to a significant reduction of cancer recurrence or mortality. The consumption of decaffeinated coffee had no impact on patients’ outcomes. Similar results have been obtained by Hu et al. [79]. Patients consuming 4 cups of coffee per day had a significantly lower risk of CRC-related as well as all-cause death ratios (HR 0.48, *p*-trend 0.003; HR 0.70, *p*-trend < 0.001, respectively). Contrary to the results from Guercio et al.’s study, the trend was significant both for caffeinated and decaffeinated coffee. The anticancer effect was dose-dependent. The assessment of correlation of coffee consumption and specific cancer-related death ratios by Gapstur et al. showed a dose-dependent inverse association of CRC-related deaths for caffeinated and decaffeinated coffee [80]. The results from those studies indicated the anticancer activity of coffee: However, according to prospective studies, coffee inhibited developed tumors rather than prevented cancer incidence and may have been an effective adjuvant during chemotherapy.

Table 1 presents the calculated risk coefficients for colorectal cancer for coffee consumption and the level of the model’s significance for reviewed studies. Please note that the groups in each study were adjusted for specific features. Some information was provided in this article for the sake of legibility of the table. Check the original manuscripts for more information.

The results of case–control and cohort studies have been compared and summarized in several meta-analyses. A meta-analysis of case control studies by Galeone et al. revealed a significant inverse association between coffee consumption and the risk of CRC (OR 0.83, *p* < 0.001), as well as subsite colon and rectal cancers: However, odds ratios were close to 1 (OR 0.93, *p* < 0.001; OR 0.98, *p* < 0.001, respectively) [81]. A more specific meta-analysis was conducted by Horisaki et al. The authors compared cohort and case–control studies conducted on a Japanese population and found that pooled CRC risk increased with coffee consumption in cohort studies and decreased in case–control studies: However, none of the associations were statistically significant [82]. Another meta-analysis of Japanese cohort studies was performed by Kashino et al. No association for CRC was found in men and women: However, colon cancer risk had a significant inverse association with the amount of coffee consumed by women [83]. Tian et al. prepared a meta-analysis of both case-control and cohort studies with dose-response analysis. The results of this study showed a significant nonlinear inverse association between coffee consumption and CRC risk in case–control studies, with reduced risk of CRC in patients consuming at least 4 cups of coffee (OR 0.89, *p* < 0.01) and an insignificant nonlinear inverse relationship in cohort studies [84]. Similar results were obtained in a study by Li et al. Combined results of case–control studies showed a significant inverse correlation of coffee consumption and colorectal and colon cancer risk in highest versus lowest or no coffee consumption (OR/RR 0.85, *p* < 0.001; OR/RR 0.79, *p* < 0.001, respectively), while an inverse correlation in cohort studies was marginal and insignificant [2]. A meta-analysis of prospective cohort studies performed by Je et al. showed no significant difference between patients with high and low consumption of coffee and the risk of colorectal, colon, and rectal cancer (RR 0.91, *p* = 0.73; RR 0.90, *p* = 0.43; RR 0.98, *p =* 0.70, respectively) [85]. The dose-response relation was also calculated in a meta-analysis of prospective cohort studies by Gan et al. The authors found a nonlinear inverse association that was significant for consumption of more than 5 cups of coffee per day [86]. The consumption of 4 cups per day reduced the risk of CRC by 7%, whereas the inverse association was stronger at higher levels of coffee intake. A meta-analysis performed within the Continuous Update Project, analyzing research on cancer prevention and survival, performed by Norat et al., found no significant associations between coffee consumption and the risk of colorectal, colon, and rectal cancer in pooled and stratified groups [87].

There has been no clear association between coffee consumption and the risk of colorectal and subsite-related cancers in case–control studies, while prospective cohort studies have been rather unambiguous in that respect and have shown no relationship. Interestingly, almost all inverse and only a few positive associations of CRC risk and coffee consumption have been statistically significant. Also, the results of meta-analyses have been in accordance with case–control and cohort studies. Additionally, a distinction between caffeinated and decaffeinated coffee has been revealed, with a significant inverse correlation for decaffeinated coffee. This may indicate that caffeine is not the most crucial bioactive compound of coffee in preventing colorectal cancer.

## 3. Caffeine

Caffeine (1,3,7-trimethylxanthine) is a purine alkaloid. Its content in roasted robusta beans is twice as high as in *Arabica* (2.5% vs. 1.2% dry weight) [51]. Stavric et al., Lelo et al., and Gillbert et al. have shown that caffeine content in a cup of coffee varies between 84 mg and 112 mg [88,89,90]. Caffeine content is strongly correlated with the method of coffee preparation. A cup of drip or percolated coffee contains 84 or 82 mg of caffeine, respectively. A single shot of espresso may contain 40–116 mg of caffeine [91]. The caffeine level in plasma in people drinking up to 6 cups of coffee varies and can reach 2–6 mg/L [89]. On the other hand, Elmenhorst et al. have shown that the consumption of 3 cups of coffee leads to caffeine levels in plasma equal to 8–10 mg/L [92]. The main problem of correlation between epidemiological and in vitro studies is the frequent occurrence of using high pharmacological rather than physiological doses of caffeine in the in vitro studies [93].

Caffeine is an antagonist of phosphodiesterases [94,95] and adenosine receptors A_1_, A_2A_, and A_2B_ [96,97], which has the effect of stimulating the central nervous system by increasing the release of dopamine, noradrenaline, and glutamate [98,99]. Moreover, it may accelerate heart rate: It also increases systolic and diastolic blood pressure and dilates blood vessels [95,100,101,102,103,104]. It also may reduce myocardial blood flow by inhibiting adenosine receptors in blood vessels [104], and reduce cerebral blood flow as well [105]. An in vitro study by Mitani et al. showed that caffeine may suppress lipid accumulation in adipocytes by inhibiting the secretion of inflammatory cytokines [106].

Caffeine is metabolized in the liver by CYP1A2, an isoform of cytochrome P450, and N-acetyltransferase 2 (NAT2) [9]. The activity of CYP1A2 and NAT2 varies widely between subjects and depends on genetic background (single nucleotide polymorphisms (SNPs)) and environmental factors such as ethnicity and diet [107,108,109]. Smoking has the strongest impact on CYP1A2 activity [110,111], however. The first products of metabolism are paraxanthine, theobromine, and theophylline [112], but the final major metabolites are 1-methylxanthine, 1-methylurate, and 5-acetylamino-6-formylamino-3-methyluracil [113]. Just a small amount of ingested caffeine (about 0.5–2%) is excreted unchanged with urine. The highest level of caffeine in plasma occurs 15 to 60 min after intake [113], and its half-life varies between 2 and 12 h [113,114]. The half-life of caffeine is also affected by such factors as cigarette smoking, sex, age, use of oral contraceptives, and pregnancy. For example, during pregnancy the half-life of caffeine increases [113,114,115], whereas smoking accelerates the metabolism of caffeine, leading to its shortened half-life [113]. Moreover, diet may also affect caffeine metabolism. The consumption of carrots, parsnips, dill, celery, and parsley slows down caffeine metabolism, in contrast to brassica vegetables such as broccoli, cauliflower, radish sprouts, and cabbage [113]. It must also be remembered that caffeine biotransformation depends on variations in CYP1A2 enzyme activity [116,117].

### The Impact of Caffeine on Cancer Cell Progression In Vitro

Saito et al. demonstrated that caffeine increased cell growth inhibition and apoptosis rate in HCT116, SW480, and DLD-1 human CRC cell lines transfected by the *PTEN* suppressor gene [118]. Treatment with combinations of Ad-*PTEN* (adenovirus mediated *PTEN* gene transfer) and caffeine selectively induced the synergistic suppression of cell growth and apoptosis in colorectal cancer cells, but not in normal cells, through abrogation of G_2_/M arrest, downregulation of the Akt pathway, and modulation of the p44/42MAPK pathway. *PTEN* (phosphatase and tensin homolog deleted on chromosome ten) is a suppressor gene (locus 10q23.31). In many studies, *PTEN* has shown the ability to inhibit cell proliferation via apoptosis boost and increase cycle cell arrest at the G1 phase [115,116,117]. In the above-mentioned experiment, the authors observed a significant increase in cells in the sub-G0/G1 phase, particularly in HCT116 p53 (+/+) cells treated by a combination of caffeine and Ad-*PTEN* (13.05% of apoptotic cells in combined treatment, compared to 7.96% in cells transduced solely with Ad-*PTEN*) [118]. Merighi et al. have shown that caffeine significantly inhibited adenosine-induced HIF-1alpha protein accumulation in colorectal cancer cells cultured in hypoxic conditions [93]. Moreover, the pretreatment of cells with caffeine significantly reduced adenosine-induced VEGF promoter activity and VEGF and IL-8 expression. The mechanism of caffeine action seemed to involve the inhibition of the extracellular signal-regulated kinase 1/2 (ERK1/2), p38, and Akt, leading to a marked decrease in adenosine-induced HIF-1α accumulation, VEGF transcriptional activation, and VEGF and IL-8 protein accumulation. In this case, caffeine treatment resulted in a decrease in proliferation and migration potential in HT29 colorectal carcinoma cells. Choi et al. investigated the sensitivity of RKO colorectal carcinoma to radiotherapy in vitro [119]. They showed that caffeine at 3 mM concentration increased the sensitivity to radiation in RKO cells. They also demonstrated that caffeine reduced the activation of ATM kinase induced by radiation. As a result, it reduced the accumulation of cells in the G2 phase. Interestingly, Mhaidat et al. assessed the impact of 20 μM of caffeine administered to Colo205 colorectal cancer cells in vitro and their influence on paclitaxel-induced apoptosis [120]. Apparently, they found no difference in apoptosis when compared to the untreated control, together with induction of the MEK/ERK kinase survival pathway. The addition of caffeine to cells treated with paclitaxel (chemotherapeutic drug) changed the induced apoptosis from ~35% apoptotic cells after paclitaxel treatment to ~18% only, with paclitaxel and caffeine administration applied simultaneously. The administration of caffeine only increased the level of anti-apoptotic Mcl-1 and GRP78 proteins: However, the anti-apoptotic mechanism was inefficient when the authors pretreated the cells with chemical MEK inhibitor.

In vitro research may not fully reflect more complex relationships in organisms due to using high concentrations of compounds that cannot be physiologically achieved in human body. The intensity of in vitro research concerning caffeine in colorectal cancer has declined in recent years in favor of epidemiological studies.

## 4. Caffeic Acid

CA is one of the metabolites of chlorogenic acid (CGA) [121,122]. It is sometimes mistaken for caffeine, although its similarity ends in the name and occurrence. CA and CGA are nonflavonoid catecholic compounds [123] and have strong antioxidant properties [121,124]. CA occurs mostly in coffee beans, vegetables, and fruits [125,126], and 95% of caffeic acid is absorbed in the small intestine [127]. CA has been shown to possess anti-inflammatory, anticancerogenic, and enzyme-inhibiting properties (e.g., lipoxygenases, cyclooxygenases) [126]. The concentration of caffeic acid in coffee depends on the content of chlorogenic acids, and ranges from 96 to 1236 mg per average cup of coffee [128]. Monteiro et al. and Nardini et al. have reported mean concentrations of 1.6 μmol/L after 1.4 h after consumption and 20.9 ng/mL after 1 h, respectively [129,130].

Murphy et al. have analyzed pre-diagnostic concentrations of caffeic acid in serum [131]. There was no difference between patients diagnosed with CRC and healthy controls (median 430 vs. median 427 nmol/L, respectively, *p* = 0.93). The multivariable model demonstrated no relationship between CA concentration in serum and colon cancer risk. CA and CGA also have the ability to inhibit H_2_O_2_- and TNF-α-induced IL-8 production in human colorectal Caco-2 cell lines [132].

### The Impact of CA on Cancer Cell Progression In Vitro

Murad et al. have determined the influence of different concentrations and exposure time on caffeic acid (CA) and 5-Caffeoylquinic acid on the viability and apoptosis of human colon carcinoma cells HT-29 [133]. A maximum uptake of CA was observed after 2 h of incubation. Inhibition of growth depended on different exposure times and concentration of CA: 24 h of exposition to 20, 40, and 80 μM caused 20–30% growth inhibition, 48 h of exposition on a 1.25–20 μM range caused ~25% growth inhibition, while 40 and 80 μM of CA caused ~40% inhibition. About 20–25% of inhibition was observed for 1.25–5 μM of CA and about 60% for 10–80 μM after 72 h of exposure. A whole range of concentrations caused 50–65% growth inhibition after 96 h incubation: However, the effect was not linear.

Interestingly, the highest inhibition during 96 h exposure was observed for 1.25 and 2.5 μM of CA, whereas for shorter times it was 20–80 μM. In addition, 48 h treatment with 5 μM of CA increased the number of cells in the G0/G1 phase, with no changes in G2/M, while 10 μM caused both an increase in the G0/G1 phase and a decrease in G2/M. Oleaga et al. found changes in the expression of STAT5B and ATF-2 in human carcinoma cells HT29 exposed to CA [134]. STAT5B overexpression and decreased levels of ATF-2 levels were confirmed in tested cells treated with CA, as compared to the control, untreated cells [135,136]. STAT5B plays a crucial role in colorectal cancer, taking part in cell growth, cell cycle progression, and apoptosis of CRC cells [137]. Inhibition of STAT5 induces G1 cell cycle arrest, but its overexpression may result in increased apoptosis [137,138]. Decreased levels of ATF-2 may have an anticancer action, as this protein is correlated with invasion, migration, proliferation, and DNA-damaging agents resistance in human breast cancer cells in vitro [134].

Sang Joon Lee et al. coupled CA with chitosan [139]. Chitosan itself, like caffeic acid, reveals antitumor activity [140]. Chito-CA conjugates dose-dependently reduced the viability of CT26 tumor cells and induced apoptosis and necrosis in cultures by means of an unknown mechanism. Furthermore, chito-CA conjugates also decreased the invasion of tested cells in in vitro conditions [139]. The inhibitory effect of CA on cancer cells and its mechanisms has been reported, but those observations were made in cells other than colorectal cancer ones (leukemia), as well as in normal human umbilical vein endothelial cells (HUVECs) [141].

## 5. Chlorogenic Acids

Chlorogenic acids are esters composed of quinic acid and one to four residues of certain trans-cinnamic acids. They belong to a group of plant polyphenols widely present in the human diet: Fruits, vegetables, coffee, tea, wine, and propolis [142,143]. Caffeoylquinic acid may theoretically have four isomers, but only three are present in plants and contain chlorogenic acid: 3-*O*-caffeoylquinic acid (3-CQA), neochlorogenic acid (5-*O*-caffeoylquinic acid, 5-CQA), or cryptochlorogenic acid (4-*O*-caffeoylquinic acid, 4-CQA). The most popular isomer, 5-CQA, is called chlorogenic acid (PubChem CID: 1794427CGA, CGA). It is an ester composed of caffeic acid and (−)-quinic acid [144]. The major source of chlorogenic acid in human diet is coffee, and its ingestion varies, depending on daily coffee consumption: Coffee drinkers ingest 0.5–1 g, while coffee abstainers ingest less than 100 mg of it per day [134]. Olthof et al. determined that about one-third of chlorogenic acid is absorbed intestinally [127], while the remaining two-thirds reach the colon and are most likely hydrolyzed to caffeic acid by colon cancer cells [145,146]. Epidemiological studies performed by Farah et al., Stalmach et al., and Renouf et al. have confirmed that caffeic acid isomers 3-CQA, 4-CQA, and 5-CQA are present in plasma and urine after consumption of coffee [147,148,149,150]. The concentration of chlorogenic acids in plasma and urine depends on ingested doses of coffee, but the results have been inconsistent. Some researchers have reported high amounts of intact 3-, 4-, and 5-CQA [129,147], while others have reported their absence [130,148]. Monteiro et al. showed the mean total CQA concentration to have been at 4.89 μmol/L for 4 h after consumption of coffee, while Farah et al. showed the mean concentration of total CQA to have been at 8.2 μmol/L for 3 h after consumption of capsulated decaffeinated green coffee extract [129,147]. CQA concentrations in serum after drinking green coffee bean extract were also measured by Matsui et al. The mean concentrations of 3-, 4-, and 5-CQA after 1 h from ingestion were 3.91, 6.81, and 7.39 ng/mL, respectively [151].

It is still disputed whether chlorogenic acid prevents or induces DNA damage. Some researchers have claimed it has a protective effect against free radical induced DNA oxidation in human colon HT29 and liver HepG2 cancer cell lines, human red blood cells, and human blood lymphocytes [152,153,154], while Xu et al. has confirmed antioxidant and DNA-protective action of 5-CQA and its isomers 3-CQA and 4-CQA [155]. On the other hand, a study by Burgos-Morón revealed that micromolar concentrations of chlorogenic acid generated DNA damage [156]. DNA damage was also observed by Li and Trush, Zheng et al., and Fan et al., but its damaging potential was quite low and increased in the presence of Cu(II) ions [157,158,159]. DNA damage, especially double-strand breaks and interstrand crosslinks, may be caused by many factors, such as the presence of reactive oxygen species and chemical compounds, or it may occur during physiological processes taking place in cells [160,161]. Both unrepaired and abnormally repaired double-strand breaks may cause rearrangements of chromosomes and loss of heterozygosity, which may result in cancer or cell death. Double-strand breaks may be repaired by nonhomologous end joining (NHEJ) and its variants or homologous recombination. Unregulated homologous recombination is one potential cancer promoter. Mutations of many homologous recombination genes have been reported in different cancers, including a mutation of the *RAD54* gene in colon cancer, which leads to the conclusion that flawed DNA damage response and homologous recombination defects cause cancer and are common in cancer cells [160,162]. The inhibition of proteins involved in DNA repair is used in anticancer therapy [163].

### The Impact of Chlorogenic Acid and Its Isomers on Cancer Cell Progression In Vitro

The number of research studies concerning CGA in colorectal cancer in vitro is still limited. Hou et al. assessed the influence of CGA on the viability of two human colorectal cancer cell lines, HCT116 and HT29, and CGA-induced ROS production in vitro [164]. They found that cell viability decreased in a dose-dependent manner for both tested cell lines. Cell viability was measured with MTT assay after 72 h exposure to CGA. There, 250 μmol/L of CGA caused the decrease of viability by ~36.0% and ~25.22% in HCT116 and HT29 cells, respectively. The highest assessed concentration, 1000 μmol/L, caused viability reduction by 50.1% and 55.2% for HCT116 and HT29 cells, respectively. The intracellular ROS level was measured using 5- (and 6-)chloromethyl-2′,7′-dichlorodihydroflurorescein diacetate (CM-H2DCFDA) and fluorescence microscopy. There, 250 μmol/L of CGA increased the ROS level by about 30% or 20%, while 1000 μmol/L increased the production of ROS by ~110% and ~90% in HCT116 and HT29 cells, respectively, when compared to the untreated control. The authors investigated cell cycle changes using PI and found that CGA induced cell arrest in phase S. Western blot analysis demonstrated the inactivation of phosphorylated ERK and increased levels of phosphorylated p53 in both lines treated with both concentrations of CGA. As the authors stated, S-phase arrest and inactivation of ERK may explain the lower viability of CGA-treated cells. Inhibition of the proliferation of the Caco-2 cell line was also found in a study performed by Wang et al. [165]. Wang et al. found an inhibition of proliferation in Caco-2 and HepG2 cells in vitro after administration of chlorogenic acid, with a median effective dose of (EC_50_) 2.8 ± 0.7 μg and 2.3 ± 1.1 μg, respectively. Sato et al. demonstrated that CGA and CA demonstrated antioxidant activity in human colon cancer cell line Caco-2 in a dose-dependent manner [121]. Antioxidant activity was measured using total antioxidant performance (TAP) and 2-methyl-6-p-methoxyphenylethynylimidazopyrazynone (MPEC) tests.

Interestingly, Xavier et al. found no change in viability or decrease of the level of phosphorylated Akt and ERK protein kinases in HCT15 and CO115 colorectal cancer cells after administration of different concentrations of CGA [166]. CGA concentrations were between 10 and 200 μmol/L, so they were lower than those in the study performed by Hou et al. [164], but partly similar as well. Viability was assessed with MTT assay, but the differences were not significant for any concentration. The level of phosphorylated Akt or ERK protein kinases was assessed with Western blot after administration of CGA (10 and 100 μmol/L), which caused no decrease of their levels in any cell line. Viability inhibition of HT-29 cells was observed by Murad et al. after 96 h of 5-CQA treatment, while shorter times of exposure (24, 48, 72 h) showed no significant differences throughout the range of concentrations (1.25–80 μM) [134]. Doses of 5 and 10 μM of 5-CQA increased the percentage of cells in the G0/G1 phase and decreased the number of cells in the G2/M phase after 48 and 96 h of exposure. The same concentrations increased the apoptosis to 1.6- and 1.8-fold after 48 h of exposure. Early apoptosis was observed in cells treated with a 5 μM dose, while late apoptotic cells were observed after treatment with a 10 μM dose.

Other studies performed in CGA-treated leukemia and hepatocellular cancer cell lines in vitro have also revealed similar results [156,167,168,169,170]. The authors stated that CGA may be a chemo-sensitizing agent during chemotherapy of selected cancers due to inactivation of ERK by overproduction of ROS.

## 6. Kahweol

Kahweol is a coffee-specific diterpene present in coffee beans and unfiltered coffee beverages [171,172]. Kahweol is a fat-soluble factor present in *Arabica coffee* beans oil, and its concentration in coffee ranges from 0.1 to 7 mg/mL [173,174]. According to a study by de Roos et al., about 70% of ingested kahweol is absorbed in humans [175]. It acts both as an adverse and chemoprotective factor [47,175]. Kahweol is known to increase the level of blood cholesterol in animal and human models [175]. On the other hand, it protects against carcinogens such as aflatoxin B_1_, 2-amino-l-methyl-6-phenylimidazo[4,5-b]-pyridine, and 2-amino-l-methyl-6-phenylimidazo[4,5-b]-pyridine [46,48,176].

### The Impact of Kahweol on Cancer Cell Progression In Vitro

Choi et al. studied the impact of different concentrations (up to 200 μM) of kahweol, caffeine, CA, and CG on human colorectal cancer cells HT-29 [177]. Interestingly, the mere exposition to kahweol decreased the viability of cells and induced cell rounding, detachment, and lactate dehydrogenase (LDH) release in a dose-dependent manner when compared to the untreated control. The authors analyzed the level of apoptosis-related proteins using Western blot analysis. Treatment with kahweol increased the expression of pro-apoptotic-activated caspase-3 and cleaved poly ADP-ribose polymerase (PARP), simultaneously decreasing the expression of anti-apoptotic Bcl-2 and phosphorylated Akt in a dose-dependent manner. Additionally, the exposure to kahweol caused decreased expression of heat shock protein 70 (HSP 70), the protein known to promote epithelial-mesenchymal transition, to be overexpressed in tumors, and to promote their growth [178,179,180]. The authors found that HSP 70 played an important role in cytotoxicity caused by kahweol. HSP 70 expression caused a significant decrease in the expression of cleaved PARP and activated caspase-3 and turned out to increase the expression of Bcl-2 and phosphorylated Akt. The viability of cells pretreated with HSP 70 inhibitor triptolide and treated with kahweol was significantly lower than the viability of those treated with kahweol only. The impact of kahweol on human colorectal cancer cells HCT116 and SW480, and on human colon normal cells CCD-18C0, was also assessed by Park et al. [181]. The cells were treated with different concentrations of kahweol, up to 50 μM. Kahweol significantly inhibited cell growth in a time- and dose-dependent manner in both lines, but the inhibition was greater in HCT116 than in SW480 cells. Interestingly, the exposition to kahweol did not affect the proliferation of normal colon cells CCD-18Co, which may indicate cancer-specific antiproliferative activity. Using Western blot, the authors found out that kahweol treatment significantly attenuated the level of cyclin D1, which is a regulatory protein in G1-to-S phase transitions. Cyclin D1 is not expressed in normal colorectal tissue, but its overexpression has been confirmed in many cancers, including colorectal carcinoma [182,183]. The occurrence of cyclin D1 in colorectal tumors is correlated with high-grade cancer with metastases to lymph nodes and deeper invasion [184]. Patients with cyclin D1-positive expression colorectal cancer have a poorer prognosis and shorter survival [184,185]. Kahweol-induced inhibition of cyclin D1 expression is presumed to be a chemoprevention for colorectal cancer. Park et al. have observed that kahweol did not affect the levels of cyclin D1 mRNA and its promoter activity [181]. Unaffected levels of mRNA and decreased protein levels indicated proteasomal degradation of cyclin D1, which was confirmed in cells pretreated with proteasome inhibitor MG132. MG132 blocked kahweol-induced decreases of cyclin D1 protein levels. The authors stated that kahweol may decrease cyclin D1 protein stability, as it significantly reduced its half-life in HCT116 cells. Further analysis of cyclin D1 proteasomal degradation included pretreating HCT116 cells with inhibitors of ERK1/2, p38, JNK, IκK-α, and inhibitors of GSK3β kinases. Kahweol-mediated cyclin D1 degradation was decreased after inhibition of ERK1/2, JNK, or GSK3β. Additionally, kahweol induced phosphorylation of ERK1/2, JNK, and GSK3β. The activation of those kinases may induce threonine-286 phosphorylation of cyclin D1 and proteasomal degradation, which follows. Unfortunately, the authors did not provide any analysis of cell cycles, where the role of cyclin D1 is crucial. Park et al. conducted other research on human colorectal cancer cell lines HCT116, SW480, LoVo, and HT-29 treated with kahweol [186]. In that research, they analyzed the impact of kahweol on apoptosis. The administration of 50 μM of kahweol for 24 h significantly induced the cleaved PARP in all cell lines, which may indicate that kahweol may induce apoptosis. Using Western blot, the authors assessed the level of activating transcription factor 3 (ATF3), the protein known to act as a tumor suppressor in colorectal cancer. Its expression may lead to downregulation of the expression of MMP-2 [187] and cyclin D1, suppression of Ras-mediated tumorigenesis [188], and enhanced activation of the p53 protein [189]. The exposure to different concentrations of kahweol generated dose-dependent increases of ATF3 in HCT116 cells, while the administration of 50 μM significantly increased ATF3 levels in all cell lines, when compared to the untreated control. Using RT-PCR, the authors measured the level of mRNA of ATF3 and found that the kahweol-induced increase of ATF3 mRNA levels was similar to kahweol-induced ATF3 protein levels. The authors pretreated CHT116 cells with inhibitors of ERK1/2, p38, JNK, and GSK3β. It was only the inhibition of ERK1/2 and GSK3β kinases that lowered the kahweol-induced ATF3 expression, which points out their impact on ATF3 expression.

It is important to remember that none of the compounds investigated in in vitro studies have been involved in epidemiological research.

Table 2 and Figure 2 summarize the current state of knowledge about the impact of caffeine, CA, CGAs, and kahweol on colorectal cancer cells in vitro with corresponding mechanisms.

## 7. Conclusions

The content of bioactive compounds in coffee depends on many conditions, such as coffee species, the plantation where the coffee was grown, bean roasting time and procedure, grinding, and preparation method [32,33,34,35].

The impact of coffee on colorectal cancer has been investigated in many epidemiological studies. Statistically significant positive and inverse associations were found for colorectal and subsite-related cancers in various case–control research papers, but some results were significant in pooled groups, while others only after adjusting for sex. Significant inverse associations for decaffeinated coffee indicates that caffeine has a lower impact on CRC in vivo than it was reported in the in vitro studies. On the other hand, prospective cohort studies found no correlation.

The results of the presented in vitro studies revealed that selected compounds, namely caffeine, CA, CGAs, and kahweol, turned out to diminish the growth of human colorectal cancer cell lines, mainly by cell phase arrest and apoptosis boost. The in vitro setting of the studies must be kept in mind, as it may not have fully reflected the more complex relationships in the organism as a whole. The results may not have been identical to those observed in living subjects due to lack of physiological processes such as accumulation in tissues, the influence of individual human gut microbiota, or intolerance to certain foods. Additionally, concentrations of bioactive compounds or exposure times in the in vitro studies may have been inadequate for in vivo conditions. The results obtained from epidemiological studies of eating habits such as coffee consumption and colorectal cancer risk have not been consistent, possibly because of attenuation of associations due to measurement errors in dietary exposure determination.

## Figures and Tables

**Figure 1 molecules-23-03309-f001:**
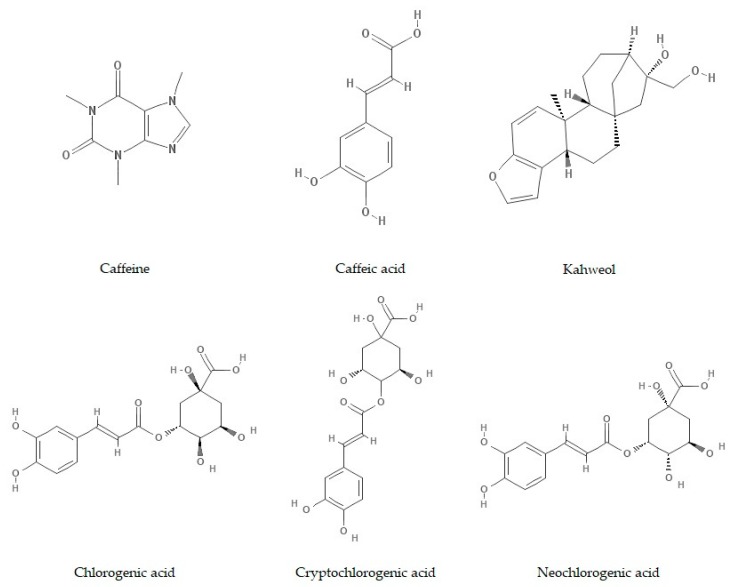
Chemical structure of caffeine, caffeic acid (CA), kahweol, and chlorogenic acids (CGAs) (adapted from Pubchem).

**Figure 2 molecules-23-03309-f002:**
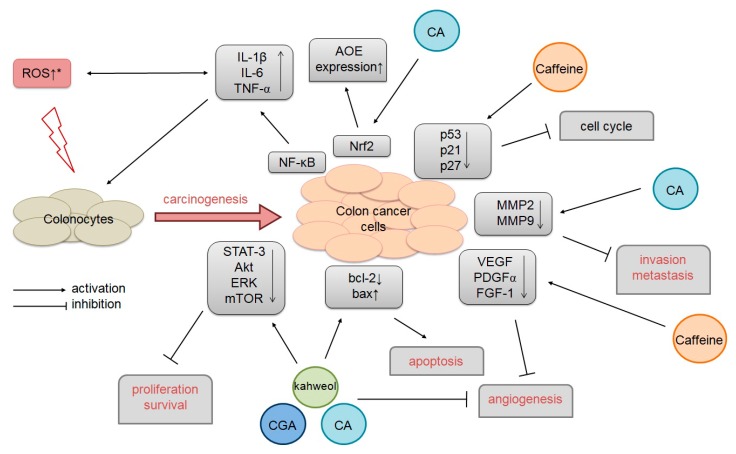
The potential mechanism of caffeine and nonflavonoid catecholic compound influence on colorectal cancer. (*) ROS leads to spontaneous goblet cell dysfunction, impaired mucosal barrier function, and inflammation, and exacerbates the development of pathology (such as cancer). CA: Caffeic acid; CGA: Chlorogenic acid.

**Table 1 molecules-23-03309-t001:** The association between colorectal cancer risk and coffee consumption with risk coefficients for investigated coffee intake. Case–control and prospective cohort studies are distinguished with symbols.

Coffee Type	Dose	Risk Coefficient	*p*-Trend	Reference
**No Risk**				
Total coffee	<100 mL/day	HR 1.04	0.58	Dik et al. [9] *
	230 mL /day	HR 1.06		
	450 mL /day	HR 0.99		
	>625 mL/day	HR 1.06		
Caffeinated coffee	70 mL/day	HR 1.03	0.29	
	190 mL/day	HR 1.07		
	387 mL/day	HR 1.01		
	550 mL/day	HR 1.10		
Decaffeinated coffee	6 mL/day	HR 1.10	0.74	
	82 mL/day	HR 0.96		
Total coffee	<1 cup/day	RR 0.99	0.137	Dominianni et al. [53] *
	1 cup/day	RR 0.98		
	2–3 cups/day	RR 1.07		
	≥4 cups/day	RR 1.16		
Total coffee	1–2 cups/day	HR 1.25	0.61	Bidel et al. [54] *
	3–4 cups/day	HR 1.32		
	5–6 cups/day	HR 1.14		
	7–9 cups/day	HR 1.28		
	≥10 cups/day	HR 1.03		
Total coffee	1–3 cups/day	HR 0.95	0.10	Lukic et al. [55] *
	3–7 cups/day	HR 0.83		
	>7 cups/day	HR 0.98		
Total coffee	1 cup/day	RR 0.96	0.95	Terry et al. [56] *
	2–3 cups/day	RR 0.93		
	≥4 cups/day	RR 1.04		
Total coffee	1–3 occ/day	HR 1.56	0.07	Nilsson et al. [59] *
	≥4 occ/day	HR 1.43	0.168	
Caffeinated coffee	0.5 cup/day	HR 1.05	0.6	Michels et al. [60] *
	1 cup/day	HR 0.99		
	2–3 cups/day	HR 1.02		
	4–5 cups/day	HR 0.98		
	>5 cups/day	HR 0.98		
Decaffeinated coffee	0.25 cup/day	HR 0.82	0.08	
	0.5 cup/day	HR 0.70		
	1–1.9 cups/day	HR 0.74		
	≥2 cups/day	HR 0.82		
Total coffee	occasionally	HR 1.01	0.73	Naganuma et al. [61] *
	1–2 cups/day	HR 0.86		
	≥3 cups/day	HR 1.00		
**Decreased Risk**				
Total coffee	≥1 and <2 servings/day	OR 0.78	<0.001	Schmit et al. [63] #
	≥2 and ≤2.5 servings/day	OR 0.59		
	>2.5 servings per day	OR 0.46		
Total coffee	<1 cup/week	HR 0.97	0.001	Sinha et al. [64] *
	1 cup/day	HR 1.00		
	2–3 cups/day	HR 0.98		
	4–5 cups/day	HR 0.87		
	≥6 cups/day	HR 0.80		
Caffeinated coffee	<1 cup/week	HR 0.97	0.008	
	1 cup/day	HR 0.99		
	2–3 cups/day	HR 1.01		
	4–5 cups/day	HR 0.90		
	≥6 cups/day	HR 0.83		
Decaffeinated coffee	<1 cup/week	HR 0.96	<0.001	
	1 cup/day	HR 1.01		
	2–3 cups/day	HR 0.93		
	4–5 cups/day	HR 0.79		
	≥6 cups/day	HR 0.74		
Total coffee	1–3 cups/week	OR 0.88	0.01	Wang et al. [65] #
	4–6 cups/week	OR 0.66		
	1–3 cups/day	OR 0.65		
	≥4 cups per day	OR 0.82		
Total coffee (consumed by men)	<1 cup/day	RR 0.96	0.91	Lee et al. [66] *
	1–2 cups/day	RR 0.94		
	>3 cups/day	RR 1.10		
Total coffee (consumed by women)	<1 cup/day	RR 0.92	0.42	
	1–2 cups/day	RR 1.01		
	>3 cups/day	RR 0.68		
Total coffee (consumed by men)	occasionally	HR 0.51	0.52	Sugiyama et al. [67] *
	1 cup/day	HR 0.67		
Total coffee (consumed by women)	occasionally	HR 0.74	0.02	
	1 cup/day	HR 0.26		
Total coffee	<1 cup/day	OR 0.88	0.009	Nakagawa et al. [68] #
	1–2 cups/day	OR 0.90		
	≥3 cups/day	OR 0.78		
Total coffee	1–2 cups/day	OR 0.08	0.002	Azzeh et al. [69] #
	3–5 cups/day	OR 0.25		
	>5 cups/day	OR 0.11		
**Increased Risk**				
Caffeinated coffee (consumed by women)	>0 to <4 cups/day	HR 1.15	0.04	Groessl et al. [75] *
	≥4 cups/day	HR 1.14		
Hot coffee	<1 cup/day	AOR 1.22	0.295	Green et al. [76] #
	1 cup/day	AOR 1.36		
	≥2 cups/day	AOR 1.24		
Iced coffee	<1 cup/month	AOR 0.89	0.035	
	<1 cup/week	AOR 1.64		
	≥1 cup/week	AOR 1.19		
Decaffeinated coffee	<1 cup/month	AOR 0.68	0.536	
	<1 cup/week	AOR 1.28		
	≥1 cup/week	AOR 1.14		
Total coffee (consumed by men)	1 cup/day	HR 1.11	0.03	Yamada et al. [77] *
	2–3 cups/day	HR 1.21		
	≥4 cups/day	HR 1.57		
Total coffee (consumed by women)	1 cup/day	HR 0.97	0.61	
	2–3 cups/day	HR 1.04		
	≥4 cups/day	HR 1.42		

Total coffee: Not distinguishing between caffeinated and decaffeinated coffee; HR: Hazard ratio; RR: Relative risk; OR: Odds ratio; AOR: Adjusted odds ratio; * cohort study; # case–control study.

**Table 2 molecules-23-03309-t002:** Potential mechanisms of caffeine, caffeic acid, chlorogenic acids, and kahweol on antitumor activity.

Type of Cancer	Cell Line	Chemical Form of Substances	Effect	Mechanism	Reference
Human colorectal cancer	HT29	Caffeine	Decrease in proliferation and migration potential	↓ HIF-1α, VEGF	Merighi et al. [93]
				↓ ERK1/2, p38, Akt	
Human colorectal cancer	HCT116	Caffeine	Induction of apoptosis	G2 phase arrest	Saito et al. [118]
	transduced with Ad-*PTEN*			↓ Akt kinase	
Human colorectal cancer	RKO	Caffeine	Increase in radiosensitivity of tumor cells	↓ Radiation-induced activation of ATM kinase	Choi et al. [119]
				↓ Activation of Chk2 kinase	
				↓ Accumulation of cells in G2 phase	
Human colon carcinoma	HT-29	Caffeic acid	Decrease of proliferation and viability	↑ G0/G1 phase arrest	Murad et al. [133]
		Chlorogenic acid	Increase of apoptosis;		
Mouse colorectal carcinoma	CT26	Caffeic acid conjugated with chitosan	increase in apoptosis and necrosis	Unknown	Lee et al. [139]
			Decrease in proliferation and invasion;		
			decrease in viability		
Human colorectal cancer	HCT116	Chlorogenic acid	Decrease in cell viability and proliferation	↑ S phase arrest	Hou et al. [164]
				↑ ROS production	
	HT29			↑ p53 Phosphorylation	
				↓ ERK Phosphorylation	
				↓ Tyrosinase activity	
Human colon cancer Human liver cancer	Caco-2	Chlorogenic acid	Decrease of proliferation	Unknown	Wang et al. [165]
	HepG2				
Human colorectal cancer	HT-29	Kahweol	Decrease of viability	↑ Cleaved PARP	Choi et al. [177]
				↑ caspase-3	
			Increase in apoptosis	↓ Bcl-2	
				↓ Phosphorylated Akt	
Human colorectal cancer	HCT116 SW480	Kahweol	Inhibition of cell growth and proliferation	↑ JNK Phosphorylation	Park et al. [181]
				↑ GSK3β Phosphorylation	
				Phosphorylation	
				↓ Cyclin D1protein level	
Human colorectal cancer	HCT116	Kahweol	Increase of apoptosis	↑ ATF3 protein level	Park et al. [186]
	SW480				
	LoVo				
	HT-29				

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
