# Peer review of "The Impact of Coffee and Its Selected Bioactive Compounds on the Development and Progression of Colorectal Cancer In Vivo and In Vitro"

_molecules, 2018, doi:10.3390/molecules23123309_

Round 1

Reviewer 1 Report

In the review article “Impact of coffee and its selected bioactive compounds on development and progression of colorectal cancer in vivo and in vitro” by Bułdak et. al., the authors have reviewed the impact of caffeine, caffeic acid, chlorogenic acids and kahweol  on proliferation, viability, invasiveness, and metastasis, as well as sensitivity to chemo- or radiotherapy of colorectal cancer cell lines. The manuscript is straightforward, well written and concise. The manuscript however has some minor flaws that need to be addressed before publication.

Concerns:

1.   The overall format of the manuscript, including the figures, should be corrected. Additionally, there are a many grammatical and spelling errors that need to be corrected.

2.   The section “The impact of caffeine on cancer cells progression in vitro” is too short and lacks vital information. This section needs to be expanded.

3.   There are several missing citations in the manuscript. For example, in the “Coffee and colorectal cancer in vivo” where the authors discuss DNA damage and again later when they discuss“Chlorogenic Acids”, they should include the following recent reviews on DNA damage agents and repair pathways, the key repair factors and the underlining molecular mechanisms:

a)            Choices have consequences: The nexus between DNA repair pathways and genomic instability in cancer. Clinical and Translational Medicine 2016; Dec 5(1): 45

b)            DNA damage response and cancer therapeutics through the lens of the Fanconi Anemia DNA repair pathway. Cell Communication and Signaling 2017; Oct 15(1):41

c)            Synthetic lethality in DNA repair network: A novel avenue in targeted cancer therapy and combination therapeutics. IUBMB Life 2017; Dec 69(12):929-937

4.   The reference list needs formatting.

Author Response

Dear reviewer,

We would like to thank you for important comments and remarks. Our response is attached as a separate file.

Reviewer 2 Report

The paper needs to make a clearer distinction between the relationships of coffee and colorectal cancer incidence and progression/survival. For CRC incidence, there is no relationship from cohort studies. I would remove all mention of associations found in case-control studies – recall and selection bias are major factors here. Instead, the authors should discuss meta-analysis evidence from prospective cohort studies. Recently, the World Cancer Research Fund has released the continuous update project (https://www.wcrf.org/sites/default/files/colorectal-cancer-slr.pdf). In this, they conducted a meta-analysis >12 cohort studies and found no relationship between coffee consumption and CRC risk (RR per 1 cup/day=1.00, 95% CI: 0.99-1.02).

However, emerging (but limited) evidence suggests that coffee drinking post-diagnosis may lower CRC deaths (https://www.gastrojournal.org/article/S0016-5085%2817%2936368-0/fulltext). This paper was not mentioned in the review and the distinction between the possible heterogeneous relationships between prediagnostic and postdiagnosis intakes and how they relate to CRC risk and survival should be fully discussd.

Some general comments:

·         Abstract - Coffee consumption is not a dietary pattern.

·         Introduction - Please reword sentence beginning “In this review…”

·         Throughout please state epidemiological studies rather than in vivo studies.

·         When discussing caffeic acid, this epidemiological study should be cited/discussed: https://onlinelibrary.wiley.com/doi/epdf/10.1002/ijc.31563

·         Suggest deleting first paragraph of page 3 (lines 78-88) as not relevant.

·         The authors suggest that the relationship between coffee consumption and CRC risk in vivo (epidemiological studies) is “inconclusive”. This is not the case, as stated above, null relationships have consistently been found in prospective epidemiological studies.

Author Response

(The authors gave the same response as above.)

Reviewer 3 Report

In this manuscript, the authors review the ability of coffee to influence colorectal cancer.  The authors discuss human studies and cell line studies. They mention animal studies but do not discuss any such studies. This needs to be added to the review.  Newer human studies need to be referenced and discussed. Given that there have been recent coffee reviews, additional discussions should be added to give a broader overview of coffee's effects on the body.

Issues:

1.    A recent rat study demonstrated that coffee, but not decaf or caffeine itself, provided chemoprotection against a carcinogen (Nutr Cancer 2018 PMID30362831). This suggests caffeine is not the main active component of coffee. An older mouse study demonstrated suppression of colon cancer metastasis by coffee (Carcinogenesis 2011, PMID21317303). 

2.    Ref #22 is supposed to be an animal study on coffee. Coffee is not mentioned in the article.  How many other references are incorrect?

3.    A more recent publication on humans (Oncotarget 2017 PMID27078843) did a meta-analysis of prospective cohort studies concluding that >4 cups of coffee/day decreased risk of colorectal cancer. Likewise, a study in Gastroenterology 2018 (PMID29158191) demonstrated a lower risk of CRC disease with coffee intake.

4.    Cell culture studies on coffee chemicals are not that relevant to human studies. Coffee contains hundreds of compounds, is consumed orally, and is metabolized by humans. Thus, the action of a single chemical on a cell is not that informative. The authors should provide some basis for comparing in vitro chemical concentrations to those possible in humans. A person could not achieve 3 mM caffeine in their serum by drinking coffee. Discussing human PK data (i.e. PMID 21538847) on coffee chemicals would put the high in vitro concentrations in better context. Another recent study showed that the presence of coffee metabolites in human serum was inversely associated with CRC (Am J Clin Nutr 2015, PMID25762808).

5.    In addition to coffee having direct effects on colon cells, coffee has been shown to have regulatory effects on DNA methylation in immune cells (Eur J Hum Genet 2017 PMID28198392). Further, there are several studies showing effects on circulating cytokine levels, induction of autophagy, and alterations in the gut microbiome.

6.    Table 1 is difficult to follow: cups per day, or week, or month. One cup a day would have more biological effects than one cup a month and should be weighted differently. HR, OR, and RR? How are these related?

7.    Many of the references did not contain the publication year nor the journal name.

Author Response

(The authors gave the same response as above.)

Reviewer 4 Report

This review manuscript covers the valuable topics to be reported.

But this MS cites too small number of papers published recently.

If the topics mentioned in this MS are not being published much recently, this topic is not valuable to be reported.

Therefore, before publication, an addition of papers published recently should be required.

Author Response

(The authors gave the same response as above.)

Round 2

Reviewer 2 Report

No comments

Author Response

Dear reviewer,

We would like to thank you for important comments and remarks. We prepared replies for your comments below. 

Moderate English changes required

Manuscript has been proofreaded and corrected by linguist. We have attached proper certificate.

Reviewer 3 Report

The authors have updated the discussed literature articles. 

Issues:

1.    The manuscript remains difficult to read and needs significant grammatical and editorial corrections.

2.    Table 1: To assist the reader, the authors should break this table into separate subheadings - Decreased cancer risk, no risk, and increased cancer risk based on statistically significant findings. 

3.    Lines 78-82: Discussion of spent coffee grounds is not relevant to this review unless the authors have identified people that actually consume spent grounds.

4.    Line 82 - "Coffee consumption has an ….". This should be the first sentence of the paragraph starting on line 71.

5.    Line 114: "coffee consumption and CRC risk are inconclusive". Based on the data presented by the authors, the data is conclusive. Coffee either has no effect or reduces CRC risk. This should be reworded.

6.    Line 174: "coffee associated with increased incidence risk but was not significant". If the findings were not significant, there was no increased risk. p values = 0.05 are already a questionable statistic in the general science literature. Suggesting increased/decreased risk with p trend values above this is not analytically correct.

7.    The authors routinely mention that coffee roasting and brewing methods affect chemicals in the coffee. Mention this in the introduction and the discussion only, not in every section.

8.    The authors seem to organize the review based on country of consumption. If that is the case, then paragraphs starting with lines 218 and 270 should be combined.

9.    Line 295: "Please note that groups in each study were adjusted to specific features". What does that mean?

10.Line 375: "caffeine can be observed seen in newborns". First, why would anyone study coffee consumption in newborns? Second, why would anyone reference such a study? Remove this sentence.

11.Section starting on line 380. The authors need to conclude that the high concentrations necessary to observe in vitro effects are not physiologically possible in humans.

12.Line 434: how much growth inhibition was observed? This is mentioned in other sections.

13.On line 446, authors state that Lee et al coupled CA with chitosan. On line 449, the same reference used CFA-chitosan conjugates. Did the same article use both conjugates?

14.Lines 486-494: This newly added section does not provide any relevant information about coffee and cancer risk and should be removed.

Author Response

Dear reviewer,

We would like to thank you for important comments and remarks. We prepared replies for your comments below. 

1.    The manuscript remains difficult to read and needs significant grammatical and editorial corrections.

Manuscript has been proofreaded and corrected by linguist. We have attached proper certificate.

2.    Table 1: To assist the reader, the authors should break this table into separate subheadings - Decreased cancer risk, no risk, and increased cancer risk based on statistically significant findings. 

The table has been broke into subheadings as requested.

3.    Lines 78-82: Discussion of spent coffee grounds is not relevant to this review unless the authors have identified people that actually consume spent grounds.

One of reviewer asked for discussion about impact of coffee on gut microbiota, therefore we cited this article. We are convinced that bioactive compounds are present both in spent grounds and coffee in different concentrations. However, we have removed those lines as requested.

4.    Line 82 - "Coffee consumption has an ….". This should be the first sentence of the paragraph starting on line 71.

After removing lines 78-82 this sentence will be first of this paragraph.

5.    Line 114: "coffee consumption and CRC risk are inconclusive". Based on the data presented by the authors, the data is conclusive. Coffee either has no effect or reduces CRC risk. This should be reworded.

According to cohort study by Groessl et al. and Yamada et al. coffee increases risk of CRC in men and women. We have reworded this sentence. 

6.    Line 174: "coffee associated with increased incidence risk but was not significant". If the findings were not significant, there was no increased risk. p values = 0.05 are already a questionable statistic in the general science literature. Suggesting increased/decreased risk with p trend values above this is not analytically correct.

This sentence has been removed.

7.    The authors routinely mention that coffee roasting and brewing methods affect chemicals in the coffee. Mention this in the introduction and the discussion only, not in every section.

Manuscript has been changes as requested.

8.    The authors seem to organize the review based on country of consumption. If that is the case, then paragraphs starting with lines 218 and 270 should be combined.

Order of references discussed in ‘Coffee and colorectal cancer in vivo’ part is same as order of references cited in Table 1. Research conducted in same country are highlighted as, in case of Japan, results are contrary.

9.    Line 295: "Please note that groups in each study were adjusted to specific features". What does that mean?

Authors of discussed studies adjusted groups for many factors like: body mass index, diabetes mellitus, menopausal status, hormone replacement therapy, physical activity, educational level, smoking status, baseline intake of energy from fat, energy from non-fat, consumption of alcohol, fibers, dairy products, red meat and processed meat, fruits, vegetables, number of colorectal examinations up to 3 years before the start of study, study centre and many others. We decided not to list them to maintain the manuscript easier to read but mentioned the different adjustation methods in different studies.

10. Line 375: "caffeine can be observed seen in newborns". First, why would anyone study coffee consumption in newborns? Second, why would anyone reference such a study? Remove this sentence.

The sentence has been removed.

11. Section starting on line 380. The authors need to conclude that the high concentrations necessary to observe in vitro effects are not physiologically possible in humans.

Conclusion has been added.

12.Line 434: how much growth inhibition was observed? This is mentioned in other sections.

Growth inhibition has been discussed.

13.On line 446, authors state that Lee et al coupled CA with chitosan. On line 449, the same reference used CFA-chitosan conjugates. Did the same article use both conjugates?

CFA is an abbreviation for caffeic acid in cited manuscript, while we use CA. CFA-chitosan phrase has been corrected to CA-chitosan.

14.Lines 486-494: This newly added section does not provide any relevant information about coffee and cancer risk and should be removed.

This section has been added and discussed on request of other reviewer.